# Ultrasonic Anisotropy in Composites: Effects and Applications

**Igor Solodov ***, **Yannick Bernhardt, Linus Littner and Marc Kreutzbruck**

Institute for Polymer Technology (IKT), University of Stuttgart, 70569 Stuttgart, Germany;
yannick.bernhardt@ikt.uni-stuttgart.de (Y.B.); linus.littner@ikt.uni-stuttgart.de (L.L.);
marc.kreutzbruck@ikt.uni-stuttgart.de (M.K.)
* Correspondence: igor.solodov@ikt.uni-stuttgart.de

**Abstract:** Stiffness anisotropy is a natural consequence of a fibrous structure of composite materials. The effect of anisotropy can be two-fold: it is highly desirable in some cases to assure a proper material response, while it might be even harmful for the applications based on "isotropic" composite materials. To provide a controllable flexibility in material architecture by corresponding fibre alignment, the methodologies for the precise non-destructive evaluation of elastic anisotropy and the fibre orientation are required. The tasks of monitoring the anisotropy and assessing the fibre fields in composites are analyzed by using the two types of ultrasonic waves suitable for regular plate-shaped composite profiles. In the plate wave approach, the effect of "dispersion of anisotropy" has been shown to make the wave velocity anisotropy to be a function of frequency. As a result, the in-plane velocity pattern measured at a certain frequency is affected by the difference in the wave structure, which activates different elasticity against the background of intrinsic material anisotropy. Phase velocity anisotropy and its frequency dependence provide a frequency variation of the beam steering angle for plate waves (dispersion of beam steering). In strongly anisotropic composite materials, the beam steering effect is shown to provide a strong focusing of ultrasonic energy (phonon focusing). For bulk shear waves, the orthotropic composite anisotropy causes the effect of acoustic birefringence. The birefringent acoustic field provides information on stiffness anisotropy which can be caused by internal stresses, texture, molecular or/and fibre orientation. On this basis, a simple experimental technique is developed and applied for mapping of fibre orientation in composite materials. Various modes of acoustic birefringence are analyzed and applied to assessing the fibre fields in injection moulding composites and to identify the fibre lay-ups in multiply materials. The birefringence pattern is also shown to be sensitive and applicable to characterizing impact- and mechanical stress-induced damage in composites.

**Keywords:** composite anisotropy; phonon focusing; acoustic birefringence; fibre orientation; damage in composites

## 1. Introduction

High-performance composite materials are rapidly becoming a mainstream technology and material of choice in demanding applications within the aerospace, automotive, medical, defense, wind energy, sports, and industrial sectors. According to a new study, the global composites market size is projected to grow from USD 88.0 billion in 2021 to USD 126.3 billion by 2026 [1]. A rapid growth of advanced composite materials used in safety critical applications imposes strict requirements to manufacturing reliability, quality assurance of new industrial products and existing components. This evokes the development of new methodologies for the non-destructive evaluation (NDE) of composite materials in order to provide greater sensitivity in monitoring product quality and its degradation caused by environmental factors or deviations in the manufacturing process, progression of damage, etc.

A particular benefit of fibre-reinforced composites is concerned with flexibility in a smart material design whose elastic anisotropy meets requirements of bearing multi-axial mechanical or thermal loading [2]. For continuous fibre (carbon, aramid) pre-preg materials,

the required stiffness anisotropy is obtained by a suitable orientation of fibre plies and a choice of their stacking pattern in a composite laminate. In short-fibre injection moulding composites, a local fibre orientation is less controllable but is mainly determined by a streamline flow direction which is usually quite difficult to predict or/and calculate. The recognition of an overall anisotropy pattern, particular fibre directions in a stack of plies, and characterization of stiffness anisotropy of a ply lay-up are the high-priority tasks for composite manufacturing and applications.

Ultrasonic waves are widely used for probing elastic anisotropy and the estimation of elastic moduli in anisotropic materials from single crystals to fibre-reinforced composites. These applications are based on the fact that elastic anisotropy for ultrasonic waves (longitudinal, shear or surface waves) is an intrinsic material property which is fully determined by its matrix of elastic moduli [3–5]. Since composite products are manufactured predominantly as plate-like components, conventional (bulk and surface) ultrasonic waves are narrowly applicable, particularly for thin composite plates, and the plate (guided) waves are used instead [6–8]. In this case, the effect of velocity dispersion should be (if feasible) sorted out and the conditions for inversion velocity anisotropy data to quantify material stiffness to be determined.

The approaches to be applied for inverse problem of plate wave velocity anisotropy in composite materials include direct analytical calculations [9–11] and various versions of the numerical and approximate analytical methods [12,13]. In this regard, particular questions to be answered are whether and (if yes) how plate wave anisotropy (and accompanying energy flow parameters) depends on frequency. The answers are of primary importance, e.g., in plate wave application for structural health monitoring (SHM) of composite components where the plate waves are used for collecting and conveying information in various directions over a large area of an anisotropic material [14,15].

Unlike optically anisotropic materials, where optical birefringence is a well-established field of applications, such an important parameter of ultrasonic waves as polarization is actually disregarded in anisotropy measurements. A strong effect caused by the anisotropic structure of composite materials has brought back interest in studies and applications of ultrasonic birefringence [16]. The developed applications of ultrasonic birefringence are mainly concerned with stress analysis and closely related to photo-elastic experiments: the stress-induced anisotropy results in splitting of a shear wave into two orthogonally polarized waves travelling at different velocities (acoustoelastic birefringence) [17–19]. The difference in velocities was found to be proportional to the difference in principal stresses and was applied to evaluation of external and residual stresses in constructional materials, such as wood and metals [20–22].

In this paper, the effects and applications of the two ultrasonic approaches introduced above are considered in terms of characterization of the in-plane stiffness anisotropy, local fiber directions in short- and long-fiber composites, including depth-resolved measurements and multiply lay-up materials. The features of the zero-order ($a_0$- and $s_0$-) plate wave anisotropy found by calculations and validated experimentally in laminate materials are shown to be frequency-dependent and frequency-sensitive to the lay-up structure. The inverse problem of material elastic anisotropy characterization is considered for low-frequency flexural modes with axial strain domination that activate in-plane Young's modulus stiffness. Phase velocity anisotropy and its frequency dependence provide frequency variation of the beam steering angle (dispersion of beam steering) and phonon focusing for flexural waves.

Various modes of ultrasonic birefringence are proposed and analyzed theoretically. The fibre reinforcement is shown to induce decomposition of a shear wave in a pair of partial waves of different velocities and polarizations to result in an elliptical particle motion. The amplitude and phase of the receiver output signal contain the information on the fibre direction and the degree of material reinforcement. The calculations validate a high sensitivity of different birefringence setups for quantification of the in-plane stiffness anisotropy, discerning local fibre directions and inconsistency in the manufacture of multiply composite

laminate structures that result in departure from the isotropic lay-up in composite laminates. The applications also include mapping short-fibre fields in injection moulding specimens and long-fibre in-plane undulation. An opportunity to identify the fibre lay-ups in multiply composites is shown by examining the measured velocity curves against the simulation results for reference composite structures. Damage and cracking produced in composites obviously modifies its in-plane stiffness anisotropy and affects the birefringence pattern. The inverse approach enables to apply the birefringence measurements for characterizing impact- and mechanical stress-induced damage in composites.

## 2. Plate Wave Measurements: Genuine or Deceptive Anisotropy?

### 2.1. Frequency Dispersion of Plate Wave Anisotropy

Elastic wave anisotropy is deemed to be an intrinsic material property, which is fully determined by the matrix of elastic moduli. One of the significant results of our previous study is the finding that it is not true for ultrasonic waves in plates whose anisotropy is shown to be a function of frequency [23]. The "dispersion of anisotropy" occurs for plate waves in all anisotropic materials including fiber-reinforced composites. This effect is, therefore, important for applications since the plate waves are widely used for non-destructive testing of the composite materials.

The dispersion of anisotropy was first observed in our experiments in carbon fibre-reinforced plastic (CFRP) [23] and also confirmed by numerical calculations. The calculations use the interactive computer program Disperse [11], which generates dispersion curves for plate waves in transversely isotropic materials (unidirectional (UD-) composites).

To reveal the effects of dispersion of anisotropy, the calculations of the in-plane phase velocity for the $a_0$ mode ($v_{a_0}$) as a function of azimuth angle ($\alpha$) were carried out for various values of (frequency) $\times$ (thickness) ($f \cdot D$) parameter. The results of calculations are presented in polar coordinates for a wide range of ($f \cdot D$) in Figure 1. The dispersion of anisotropy is clearly seen in Figure 1: the shapes of the $v_{a_0}(\alpha)$ curves and the velocity values change evidently as the wave frequency changes. To characterize and quantify the variation of the anisotropy patterns, we introduce the anisotropy parameter ($A$) as the ratio of the phase velocities along 0° and 90° directions. A frequency dependence of this parameter quantifies the dispersion of anisotropy. The results of calculations for the $A_{a_0}(f) = v_{a_0}^0 / v_{a_0}^{90}$ based on the data in Figure 1 are given in Figure 2. As the frequency increases, it exhibits a sharp decent from a certain "static" value ($\cong$1.95, Figure 2a, insert) to an asymptotic plateau at higher frequencies.

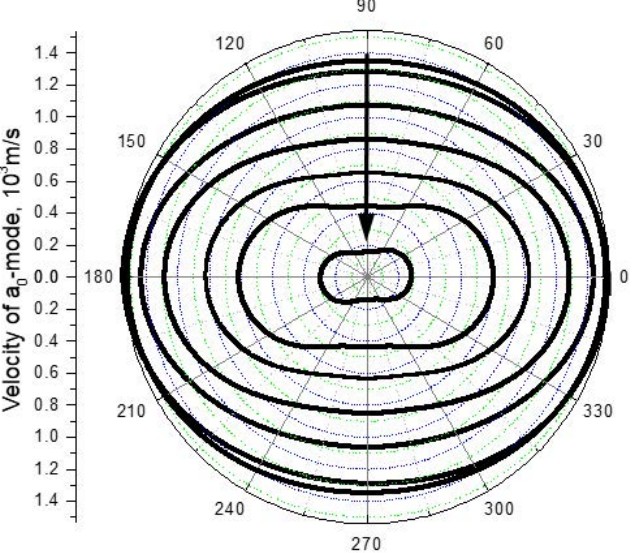

**Figure 1.** Calculations of in-plane phase velocity anisotropy. ($a_0$ mode) in UD-CFRP: consecutive values of ($f \cdot D$) parameter for the curves in the direction of arrow are: 4000, 3000, 500, 250, 175, 50 and 5.

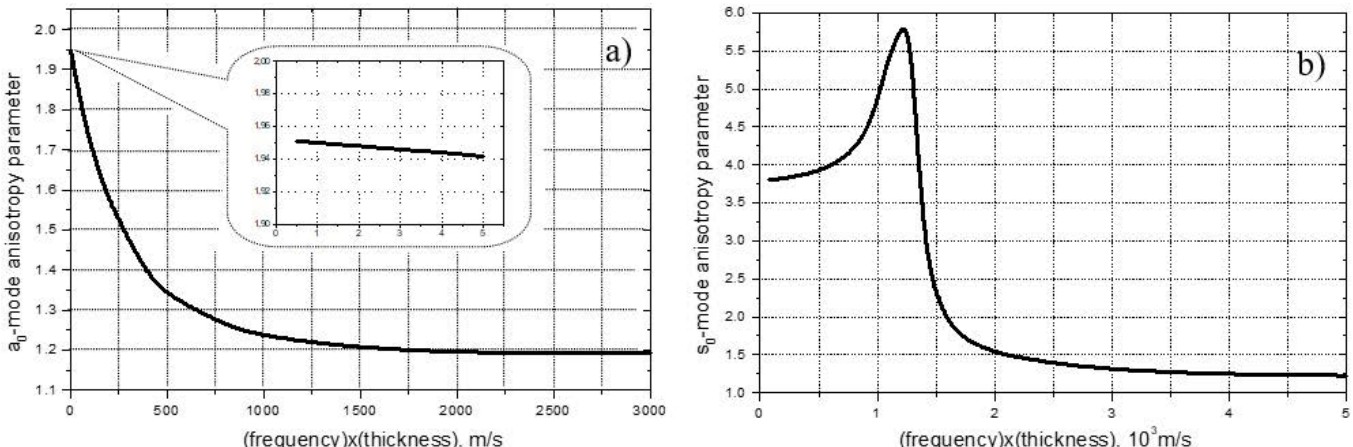

**Figure 2.** Calculations of dispersion parameters for in-plane anisotropy for $a_0$- (**a**) and $s_0$- (**b**) modes in UD-CFRP.

The estimate for the "static" value of the anisotropy parameter can also be determined from the ratio of the low-frequency velocities of $s_0$ modes [5]:

$$A_{a_0}^{static} \approx (v_{S0}^0 / v_{S0}^{90})^{1/2}. \tag{1}$$

Direct calculations of the dispersion curves for the $s_0$ modes propagating along and across the fibres in UD-CFRP yield at low frequency end $(v_{S0}^0 / v_{S0}^{90}) \approx 3.8$ (Figure 2b) so that (1) readily validates the value $A_{a_0}^{static}$ obtained above.

At high frequencies, the $a_0$ modes are gradually converted into surface acoustic waves (SAW), which determine the high-frequency end of the anisotropy dispersion curve in Figure 2a. The SAWs are non-dispersive; they produce mainly shear out-of-plane near-surface deformation and expected to have lower elastic anisotropy. This was confirmed experimentally by direct measurements of the SAW velocities along and across the fibres in UD-CFRP [24]. The measurements yield the estimate for the high-frequency limit of $a_0$-anisotropy parameter as:

$$A_{a_0}^{HF} = (v_{SAW}^0 / v_{SAW}^{90}) \cong 1.2 \pm 0.3, \tag{2}$$

which fits closely to the calculation data in Figure 2a.

Similar calculations of the anisotropy dispersion carried out for the $s_0$ modes in UD-CFRP are shown in Figure 2b. The values of $A_{s_0}$ are substantially higher than those for $a_0$ mode.

Because the symmetrical modes make good use of intrinsic composite anisotropy by producing mainly pure longitudinal deformation along and across the fibres. The low-frequency plateau in the $s_0$-velocity anisotropy is estimated as $A_{S_0}^{static} = (A_{a_0}^{static})^2 \cong 3.8$, which is in full accord with calculations in Figure 2b. At high frequencies, both $s_0$ modes are converted into SAWs; this provides a gradual decay of the $s_0$-anisotropy parameter in Figure 2 to the asymptotic plateau given by (2).

The physical reason for the dispersion of velocity anisotropy is concerned with a frequency-dependent variation in the wave structure. For example, in a low-frequency $a_0$ mode, the axial strain dominates and runs through the whole thickness of the specimen (Figure 3a). Such a strain naturally activates Young's modulus, which determines the (static) bending stiffness of a plate. As the frequency increases, the wave field is "pushed out" from the interior of the material and the axial strain is diminished while the out-of-plane shear strain enhances (Figure 3b). Material stiffness activated by the high-frequency wave is close to the out-of-plane shear modulus. The frequency-dependent structural transition, therefore, contributes to velocity dispersion of the $a_0$ mode in a selected propagation direction. As a result, the azimuth velocity pattern measured at a certain frequency is

affected by the difference in the wave structure, which activates different elasticity on the background of intrinsic material anisotropy. The contribution of the wave structure varies with frequency, thus providing the dispersion of velocity anisotropy.

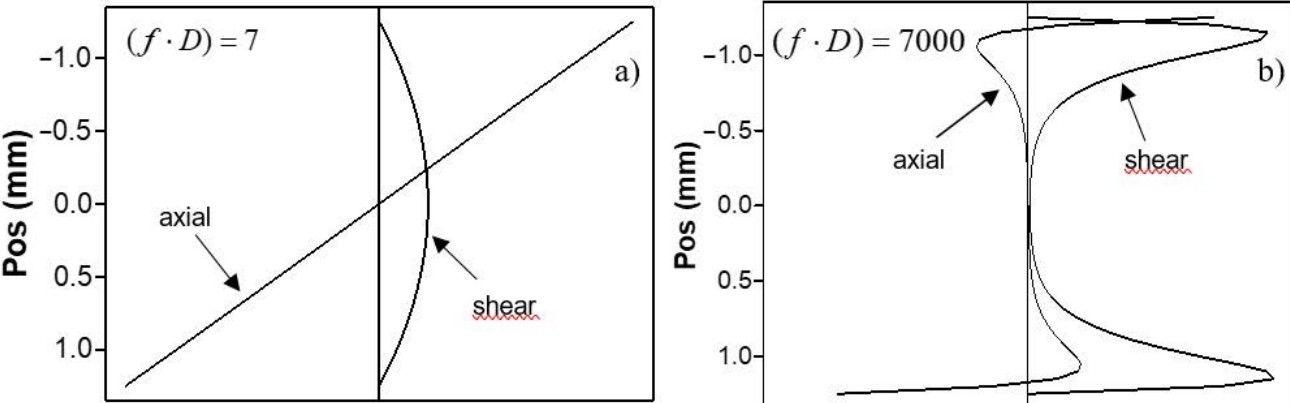

**Figure 3.** Strain distributions with depth for $a_0$ mode for different values $(f \cdot D)$ (7-(**a**), 7000-(**b**)) in 2.5 mm-thick 90°-CFRP plate.

### 2.2. Depth-Resolved Measurements of Anisotropy

As it has been shown above, the effect of plate wave structure variation causes the wave anisotropy to be frequency dependent and thus makes the direct use of the material stiffness measurements misleading. However, in some cases, it can also find beneficial applications. E.g., in multi-ply composite laminates, the plate wave velocity is determined by the contributions of all plies deformations into an overall stiffness. However, since the particle motion and the local strain produced by the wave at different frequencies change over the plate thickness, each of the contributions will depend on the degree of "activation" of a particular ply. Typical displacement patterns calculated by using the Disperse program for the $a_0$ modes of different frequencies in 0° direction of UD-CFRP are shown in Figure 4.

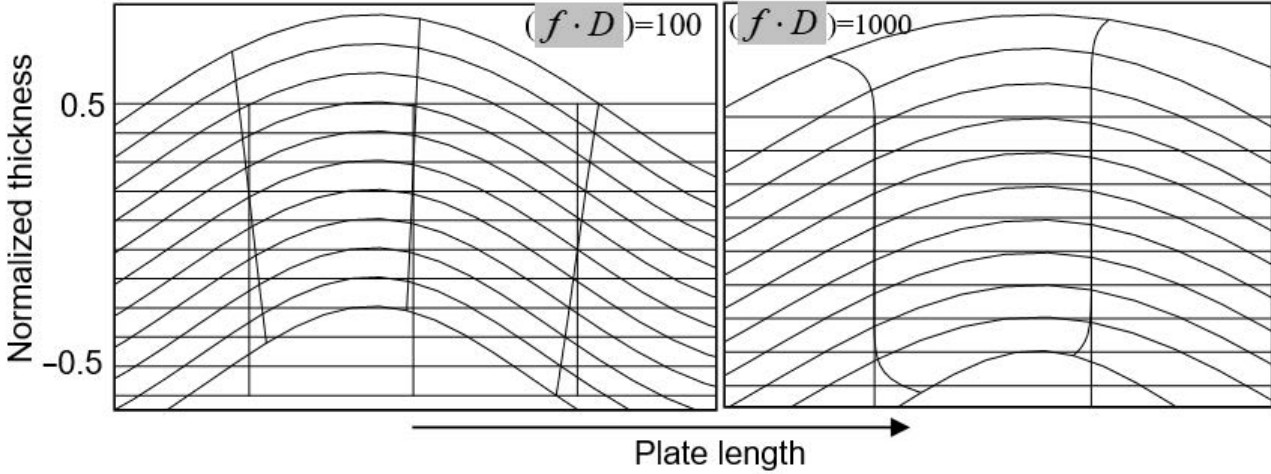

**Figure 4.** Displacement patterns for $a_0$ modes for different frequencies and various $(f \cdot D)$ in UD-CFRP.

In both cases shown in Figure 4, the in-plane displacement is maximal on the plate surface and zero in the middle plane. This provides a similar inhomogeneous depth distribution of longitudinal strain in the specimen (Figure 5). At higher frequencies, the longitudinal strain is displaced from the interior of the specimen to its outer part due to the "skin-effect". As a result, maximum contribution to bending stiffness for higher frequency $a_0$ modes is expected from the surface plies while the role of the inner plies is

diminished. This effect provides an opportunity for the depth-resolved measurements of stiffness anisotropy in laminate composites.

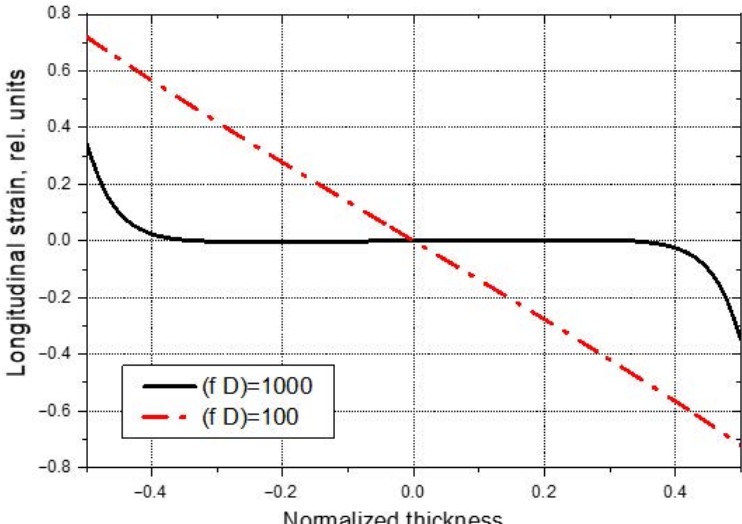

**Figure 5.** Depth distribution of longitudinal strain in UD-CFRP at two different frequencies.

To demonstrate the feasibility of the depth-resolved probing of stiffness, two specimens of UD-CFRP prepregs (HexPly M18/1 G947 UD) were manufactured. The laminate plies were made in quasi-isotropic symmetrical lay-ups [0/45/−45/90] s and [0/60/−60] s in regard to the middle plane, which is common for most applications to avoid out-of-plane bending under plane stress conditions.

A quasi-isotropic lay-up is expected to exhibit no stiffness anisotropy when the in-plane stiffness is averaged over the thickness of the sample. The lack of such "averaged" anisotropy in our specimens was confirmed by the measurements of acoustic birefringence using through the transmission of bulk shear waves [25]. Unlike the bulk wave case, the weighted averaging with the emphasis on the outer plies is expected for the flexural wave propagation. To generate and detect the 200 kHz $a_0$ modes in CFRP laminates ($f \cdot D \approx 400$), a slanted mode of air-coupled ultrasound was used in a single-sided access configuration [26]. The phase of the received signal as a function of distance between the transducers was measured for the precise evaluation of flexural wave velocity. Owing to non-contact excitation/detection, the change in the propagation direction ($\alpha$) was managed by rotation of the specimens in azimuth plane. The measurement results (Figure 6) reveal noticeable velocity anisotropy for both laminates. According to Figure 6, the in-plane velocity anisotropy is $(v_{a0}^0 - v_{a0}^{90})/v_{a0}^{90} \approx (1410/1260) - 1 \approx 12\%$ in the [0/45/−45/90] s and $(1360/1190) - 1 \approx 14\%$ in the [0/60/−60] s that makes the laminates substantially non-isotropic for plate waves.

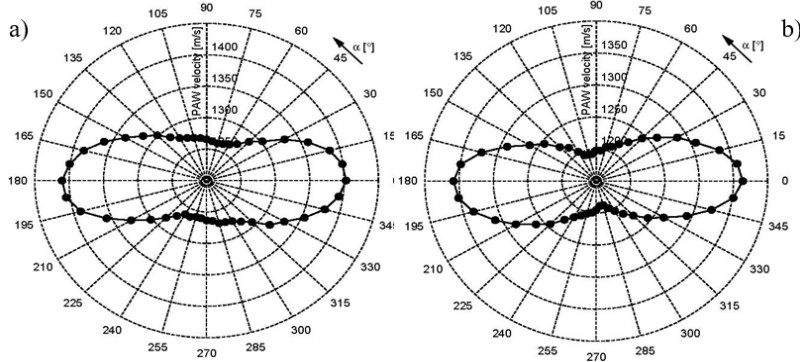

**Figure 6.** Flexural wave velocity anisotropy in quasi-isotropic CFRP laminates: [0/45/−45/90] s (**a**) and [0/60/−60] s (**b**).

### 2.3. Inversion of Plate Wave Velocity Data to Derive Material Stiffness Anisotropy

To single out material anisotropy, one has to eliminate the frequency dependent variation in the wave structure, i.e., the velocity data should be taken for the same wave structure that can be obtained by frequency variation. Figure 7 illustrates this approach for the $a_0$ modes in 2.5 mm UD-CFRP plate: the three identical in-depth strain distributions for the directions of 0°, 45° and 90° require substantially different frequencies and propagate with different velocities.

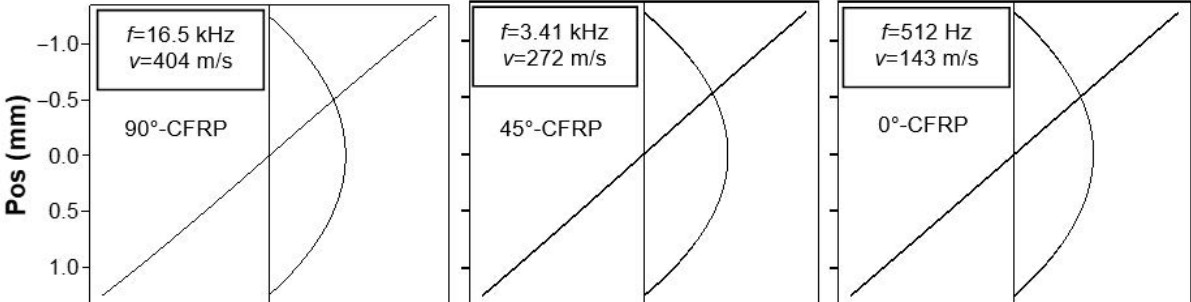

**Figure 7.** Identical strain distributions with depth calculated for $a_0$ modes in different directions of UD-CFRP at different frequencies.

Each of the wave fields correspond to initial sections of the dispersion curves where the axial strain prevails so the primary stiffness activated by the wave along the $x_i$-direction is Young's modulus $E_i$. Its contribution to the $a_0$-wave velocity is given by the relation known to be valid for $f \to 0$ [5]:

$$(v_{a_0})_i = (\pi D f)^{1/2} (E_i/3\rho(1 - \nu_{ik}\nu_{ki}))^{1/4} \tag{3}$$

where $D$ is the thickness, and in-plane Poisson's ratios $\nu_{ik}$ are involved.

For CFRP, the product of $\nu$ in (3) is $<< 1$ [27], so that Young's modulus can be found as:

$$E_i \approx [(3\rho/\pi^2 D^2)((v_{a_0})_i^4/f^2)] \tag{4}$$

To reveal Young's modulus anisotropy, the combinations of $(v_{a_0})_i$ and $f$, which activate an axial strain and form an identical wave structure similar to that given in Figure 7, were calculated for 10° steps in azimuth angle $\alpha$. The $E_i(\alpha)$ plot obtained is shown in Figure 8; a further increase in the low-frequency axial strain by reducing the frequency weakly affects the $E$ values within few % interval. The results in Figure 8 are in fair compliance with the data obtained in [27] and confirm the feasibility of the "identical wave structure" approach for the evaluation of stiffness anisotropy.

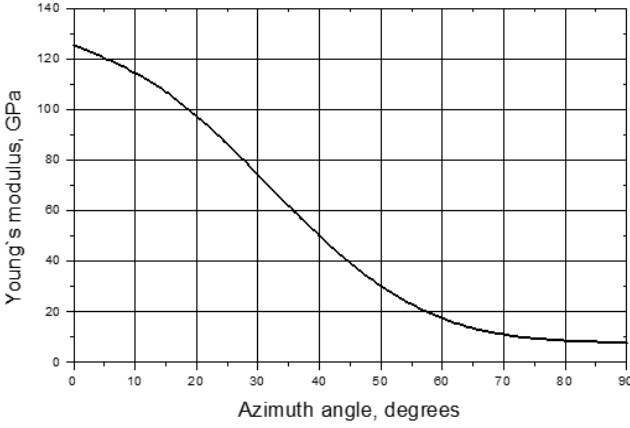

**Figure 8.** In-plane anisotropy of Young's modulus in UD-CFRP calculated by "identical wave structure" approach.

A similar approach operates well for the $s_0$ modes in a low-frequency range: according to Figure 9, the identical wave structures with axial strain dominance are found at different frequencies along and across the fibres. At (and below) these frequencies, Young's modulus anisotropy can be estimated from the relation [5]:

$$(v_{s0})_{xi} = [E_i / \rho (1 - \nu_{ik} \nu_{ki})]^{1/2} \approx (E_i / \rho)^{1/2} \tag{5}$$

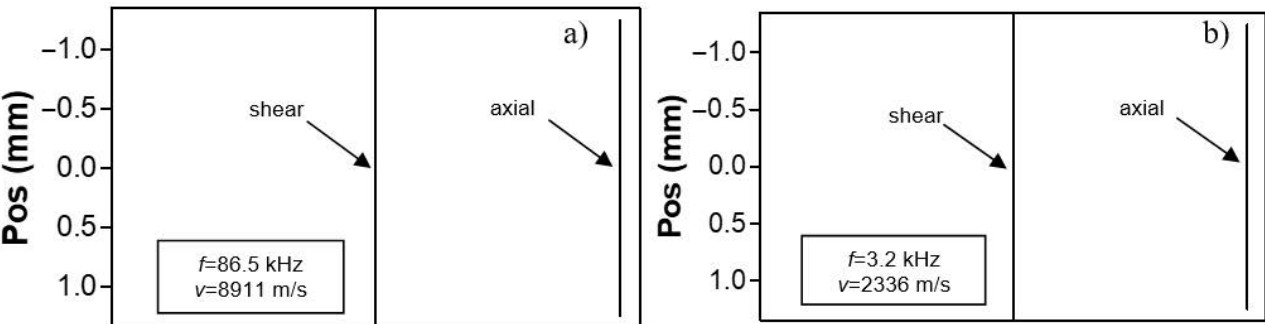

**Figure 9.** Identical strain distributions with depth calculated for $s_0$ modes in $0°$ (**a**) and $90°$ (**b**) directions of UD-CFRP at different frequencies.

By using velocity data given in Figure 9 (insets) in (5), the values of Young's moduli are calculated as: $E_0 \cong 126.6$; $E_{90} \cong 8.7$ and found to be in reasonable agreement with the data in Figure 8.

To observe the effect of anisotropy dispersion in the experiment and verify the results of the calculations, the measurements of the phase velocity for zero-order plate waves were carried out in a wide frequency range. Both the air-coupled ultrasound (ACU) [26] and wave form imaging (WIM) [28] methodologies were used for measurements of $a_0$ modes over (18–250) kHz frequency range. A broad-band excitation (150–900 kHz) by using high-frequency wave piezo-transducers was applied for velocity measurements of the $s_0$-waves. The specimen studied was epoxy cured 20-ply high-strength CFRP lay-up (weight fibre content $\approx$ 50%; thickness $D$ = 2.5 mm) consisting of (2 × 9) unidirectional (0°) carbon fibre plies and two fabric carbon fibre (±45°) plies in the middle.

From the measured velocity data in Figure 10, the frequency variation of the plate wave anisotropy is derived; the results are shown in Figure 11. A close agreement with the calculation results in Figure 2 confirms that the frequency dispersion of anisotropy is significant for both $a_0$ and $s_0$ waves in CFRP. The $s_0$-wave anisotropy is strongly affected by frequency variation due to transition from high-modulus longitudinal to low-stiffness shear deformation. For $a_0$ waves, the contribution of longitudinal deformation to low-frequency bending stiffness provides maximum anisotropy in the "static" case. The phenomenology developed enables a rapid analytical quantification of frequency dispersion of anisotropy in composite materials with known elastic coefficients.

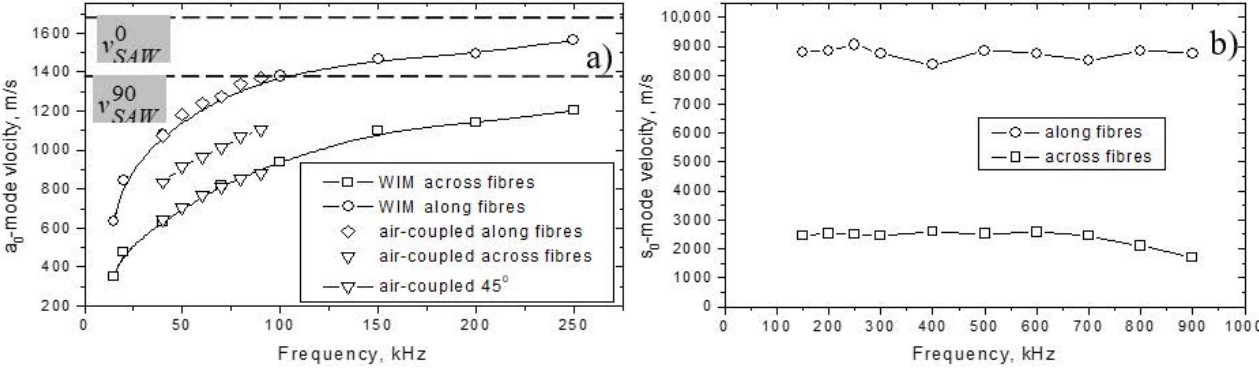

**Figure 10.** Measured velocities of $a_0$ (**a**) and $s_0$ (**b**) modes as functions of frequency in CFRP.

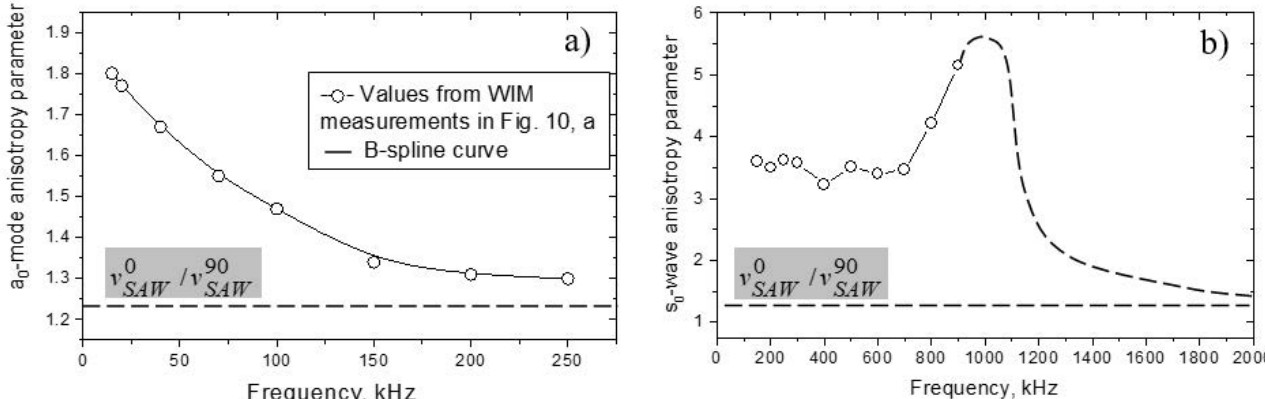

**Figure 11.** Anisotropy parameters for $a_0$ (**a**) and $s_0$ (**b**) modes derived from velocity measurements in Figure 10.

### 2.4. Anisotropy Effects in Plate Wave Propagation in Composites

The phase velocity anisotropy and its frequency dependence demonstrated above have significant implication on the plate wave propagation in strongly anisotropic composite materials. Azimuth velocity dependence is visualized readily by the WIM methodology [28]. The WIM is a rapid and intuitive technique; the problem with its application to mapping of material stiffness is that it provides an angular dependence for the group velocity. Therefore, an additional conversion from the group ($v_g$) to the phase velocity ($v_p$) pattern is required.

The relation between the group and phase velocities is easily seen from Figure 12a:

$$v_p(\theta) = v_g(\alpha) \cos \psi. \tag{6}$$

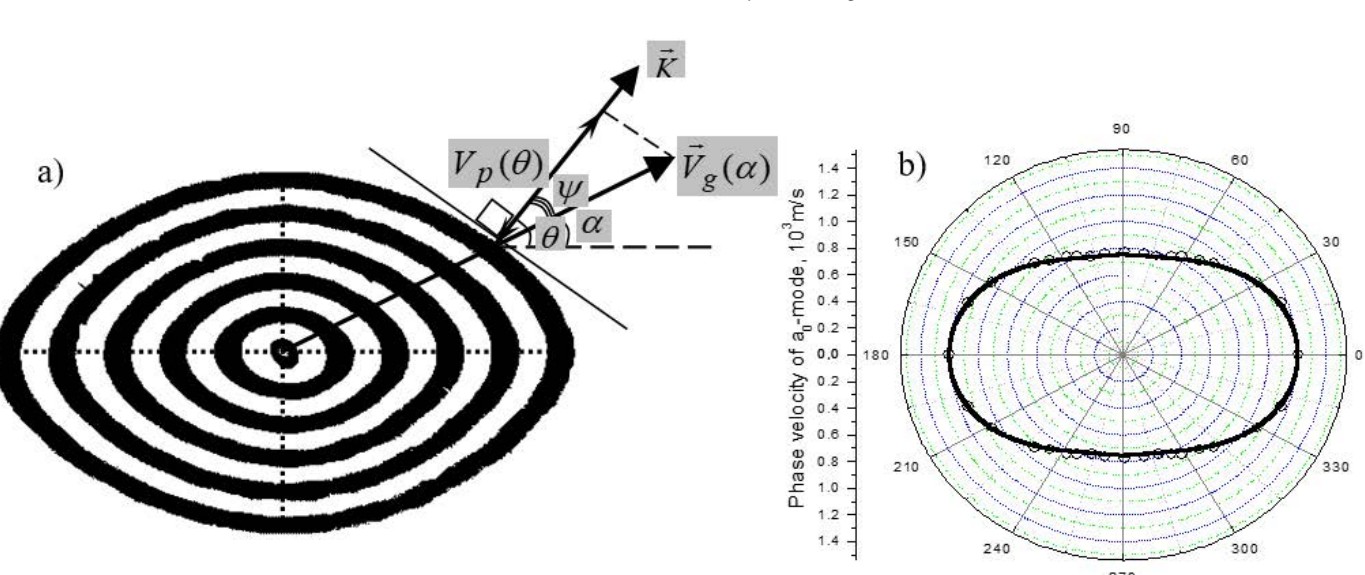

**Figure 12.** Relation between group and phase velocities for plate waves. The group velocity pattern (**a**) is measured by WIM in CFRP at $(f \cdot D) = 175$. Phase velocity pattern converted from the WIM data (**b**, circles); calculations for UD-CFRP at $(f \cdot D) = 175$ (**b**, solid line).

The orientation ($\theta$) of the wave vector $\vec{K}$ (and phase velocity) is found as a normal to a local tangent to the group velocity waveform in the direction $\alpha$. The angle $\psi$ is then determined as $\psi = \theta - \alpha$ while $v_g(\alpha)$ is measured directly from the waveform image (Figure 12a).

The conversion of the WIM data (for $a_0$ waves at $(f \cdot D) = 175$) to the phase velocity pattern based on relation (6) is illustrated in Figure 12b (circles). The solid line in this

figure shows the results of the direct calculations of the in-plane phase velocity pattern in UD-CFRP.

The data in Figure 12 demonstrate a substantial difference between the in-plane group and phase velocity patterns for $a_0$ modes. The validity of the conversion procedure is supported by an excellent agreement of the converted data with calculations using Disperse (Figure 12b). According to Figure 12a, the angle $\psi$ between the wave energy and phase propagation can be quite large to cause substantial beam steering. This effect, well-known in the acoustics of crystals [3], is basically disregarded in ultrasonic applications in composite materials. The beam steering angle is determined by a local curvature of the in-plane anisotropy curve $v_p(\alpha)$:

$$tg\psi = (1/v_p)dv_p/d\alpha. \tag{7}$$

It can also be found as $\psi = \theta - \alpha$ from the WIM experimental data. In Figure 13a, the results of calculations of $\psi$ based on (7) are compared with those derived from the WIM wave front measurements (Figure 12a) for CFRP. Both approaches reveal a strong impact of steering on the propagation of $a_0$-plate waves in the slanted directions.

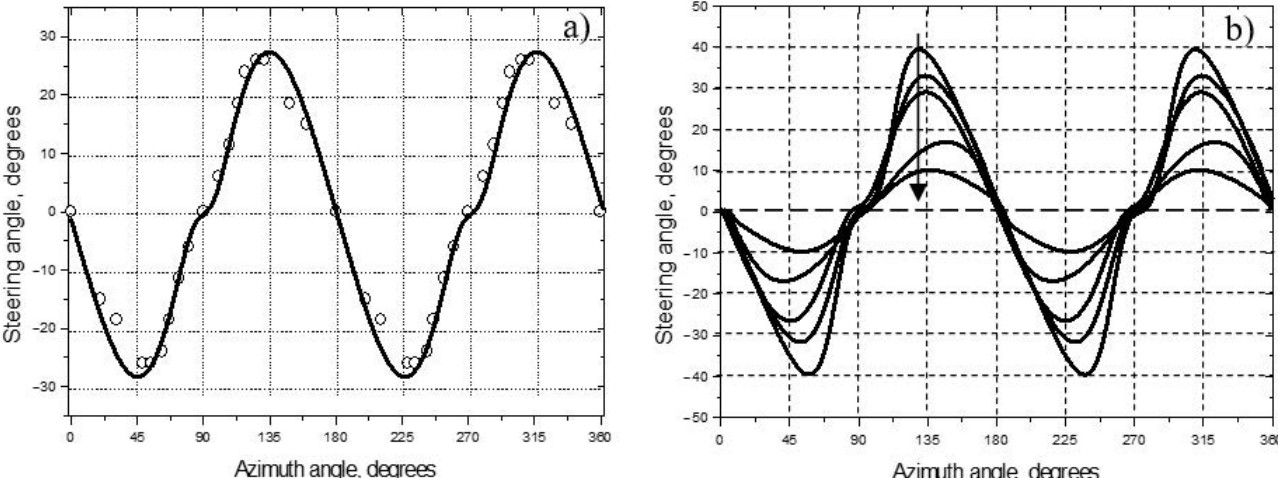

**Figure 13.** Beam steering effect for $a_0$ modes in CFRP (($f \cdot D$) = 175): (**a**) calculated in-plane beam steering anisotropy (solid line) and $\psi(\alpha)$ derived from WIM measurements (circles); (**b**) dispersion of beam steering: the values of ($f \cdot D$) parameter for the curves in the direction of arrow are: 5, 100, 175, 500, and 3000.

Unlike the case of bulk ultrasound, the dispersion of anisotropy provides frequency variation of the beam steering angle for plate waves (dispersion of beam steering). The effect is illustrated in Figure 13b: the steering angles reduce as ($f \cdot D$) parameter increases. The frequency variation of maximum $|\psi|$ values calculated predicts a very strong energy steering ($\psi \cong 40°$) for thin CFRP plates or low frequencies.

The experimental evidence for the beam steering effect in CFRP is shown in Figure 14. The $a_0$ mode is excited in 2.5 mm-thick CFRP plate with 50 kHz air-coupled transducer; the wave propagates at $(-45°)$ to the fibre direction (wave vector $\vec{K}$ in Figure 14). The wave field on the surface of the specimen is visualized with a scanning laser vibrometer. The image in Figure 14 shows that the energy flux strongly deviates from the propagation direction while the wave phase front remains normal to $\vec{K}$. The steering angle between $\vec{V}_g$ and $\vec{K}$ in Figure 14 is measured to be $\cong 32° \pm 2°$ that is in a very close agreement with the calculations in Figure 13b.

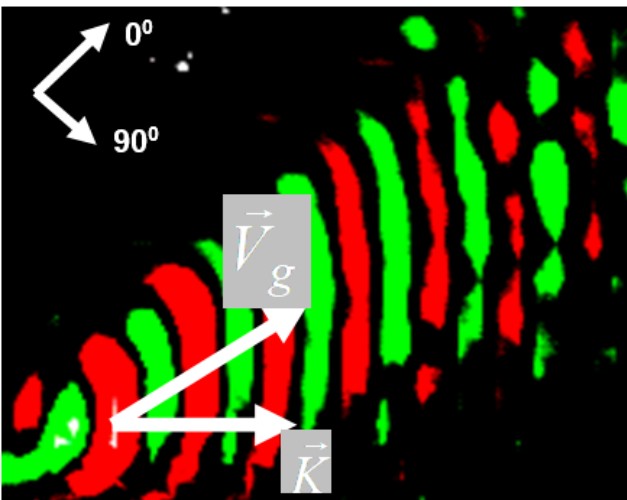

**Figure 14.** Experimental observation of beam steering in CFRP (($f \cdot D$) = 125): air-coupled $a_0$-wave propagates at ($-45°$) to fibre direction (0°).

The sign variation of the steering angle in Figure 13 indicates that for any in-plane direction of propagation, the energy flux deviates in such a way in order to stay closer to the fibre direction. Therefore, for the omnidirectional plate wave, the beam steering in CFRP should result in energy focusing in the neighbourhood of the fibre direction. This effect, known as "phonon focusing" [29], has been observed in crystalline materials [30] and is also predicted to exist in composites [31]. The energy enhancement caused by non-uniform azimuthal distribution of $\vec{V}_g$ is characterized by the focusing factor: $F = |d\theta/d\alpha|^{-1}$. Equation (7) shows that this derivative is fully determined by the in-plane phase velocity anisotropy. Since $v_p(\alpha)$ changes with frequency, the phonon focusing for $a_0$ waves will also be frequency-dependent. This is illustrated in Figure 15, where $F(\alpha)$ is calculated at different values of ($f \cdot D$) parameter. The data indicate that the phonon focusing depends strongly on frequency. The maximum focusing is expected at low ($f \cdot D$) when the material anisotropy for flexural waves increases.

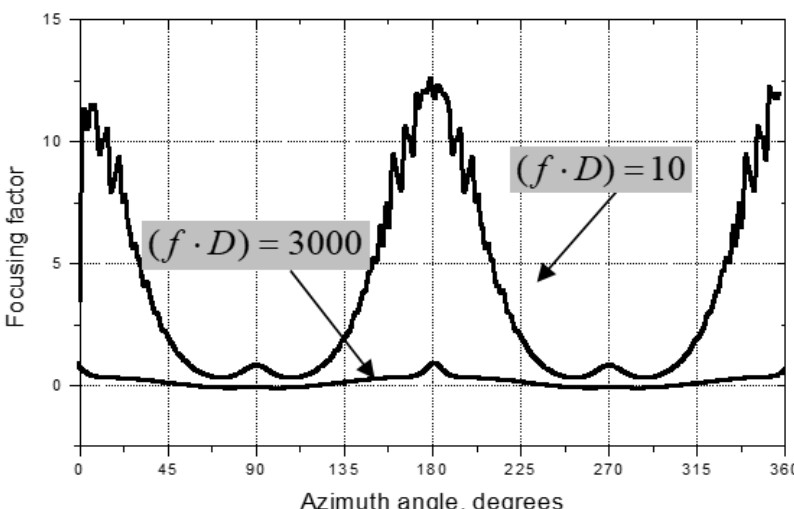

**Figure 15.** Dispersion of phonon focusing in UD-CFRP.

To experimentally observe the phonon focusing in CFRP, a small (~5 mm diameter) wide-band piezo-transducer was attached in the centre of a (300 × 200 mm²) CFRP plate and used for excitation of cylindrical $a_0$ waves. The wave field formed in the specimen was visualized on the opposite site with a laser scanning vibrometer. The images of averaged (RMS) vibration velocity distributions measured over (260 × 160 mm²) area at

two frequencies are shown in Figure 16. The focusing along the fibre direction is clearly seen in both cases. Unlike the crystalline materials, the higher wave attenuation in composites has a significant effect and makes the propagation distance much shorter for the higher-frequency waves.

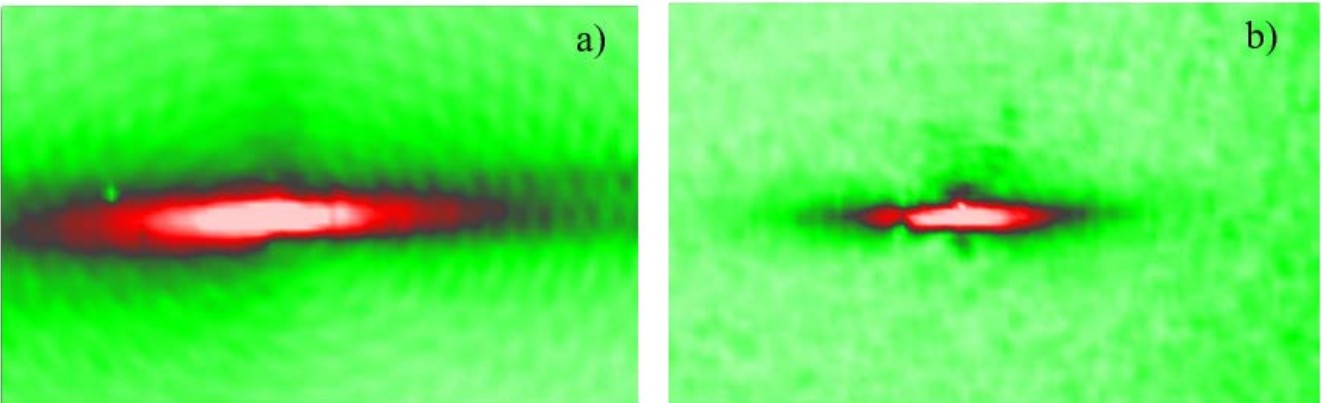

**Figure 16.** Phonon focusing of $a_0$ modes in UD-CFRP: (**a**) $(f \cdot D) = 175$; (**b**) $(f \cdot D) = 500$. The fibre direction is horizontal.

## 3. Ultrasonic Birefringence: A Remedy to Unravel a Tangle of Fibers?

### 3.1. Theoretical Background and Various Operation Modes

The reinforced materials (fibre composites) manifest different anisotropic configurations depending on the material structure and composition. The bidirectional in-plane reinforcements (Figure 17) induce three orthogonal twofold axes of symmetry and display an orthotropic anisotropy. Such anisotropy remains valid as long as both in-plane reinforcement directions are not identical. Otherwise, the material acquires a fourfold axis of symmetry (the z-axis) and upgrades to the tetragonal symmetry.

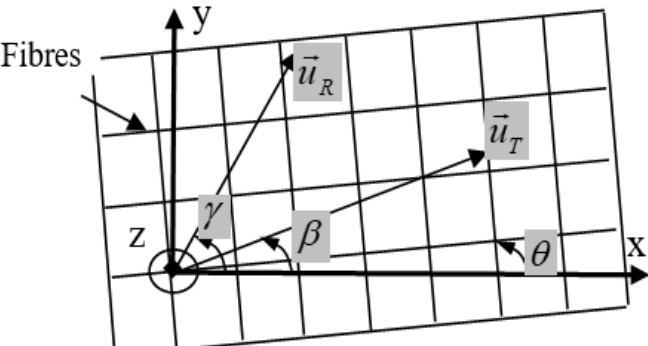

**Figure 17.** Coordinate system used in bidirectional reinforced composite material.

The impact of the fibre array on characteristics of acoustic wave propagation is obtained from the Christoffel equation [3]:

$$(c_{ijkl} n_j n_l - \rho v^2 \delta_{ik}) u_k = 0, \tag{8}$$

where $c_{ijkl}$ are elastic moduli; $n_{j,l}$ are the components of a unit vector in the direction of wave propagation; $\delta_{ik}$ is the Kronecker symbol; $\rho$ is the density of the material; $v$ is the acoustic wave velocity; and $u_k$ are the displacement vector components. A non-trivial solution of (8) requires:

$$|c_{ijkl} n_j n_l - \rho v^2 \delta_{ik}| = 0 \tag{9}$$

which determines the velocities $v_m$ ($m = 1, 2, 3$) for a given direction of propagation.

Since composite and metallic materials are often produced in a sheet-like geometry, the bulk wave propagation direction is limited by the z-axis; hence, $n_3 = 1$; $n_{1,2} = 0$. In the case $\theta = 0°$, these values and the elastic moduli matrix for orthotropic symmetry [3] are used in (8) to obtain:

$$(c_{55} - \rho v^2)u_1 = 0; \ (c_{44} - \rho v^2)u_2 = 0; \ (c_{33} - \rho v^2)u_3 = 0. \tag{10}$$

The first two relations in (10) show that the velocities for the in-plane shear waves polarized along the orthogonal fibre directions are different: $v_1 = \sqrt{c_{55}/\rho}$; $v_2 = \sqrt{c_{44}/\rho}$. The values of the elastic moduli are proportional to the number of fibres in the corresponding direction: $c_{44} > c_{55}$ in Figure 17, so that the vertical fibres form the "fast" polarization direction.

Therefore, a linearly polarized shear wave in the material (without attenuation) is decomposed into two shear waves polarized along the reinforcement directions and propagating with different velocities:

$$\vec{u}_T(z, t) = U_{T_x} \vec{e}_x \sin(\omega t - k_1 z) + U_{T_y} \vec{e}_y \sin(\omega t - k_2 z), \tag{11}$$

where $k_1$ and $k_2$ are the wave numbers for the shear waves polarized along the x- and y-axes, respectively; $\vec{e}_x$, $\vec{e}_y$ are the unit vectors along the corresponding axes. The wave amplitudes in (11) depend on the orientation of the transmitting shear wave transducer: $U_{T_x} = U_T \cos \beta$; $U_{T_y} = U_T \sin \beta$, where $\beta$ is the angle of the transmitter polarization (Figure 17) and $U_T$ is the amplitude of its displacement.

Since $k_1$ and $k_2$ are different (ultrasonic birefringence), after traversing a plate of thickness $d$, the particle displacement acquires (in a general case) an elliptical polarization:

$$\vec{u}_T(d, t) = (U_T \cos \beta) \vec{e}_x \sin \omega t + (U_T \sin \beta) \vec{e}_y \sin(\omega t + \Delta\varphi), \tag{12}$$

where the phase shift is:

$$\Delta\varphi = d(k_1 - k_2). \tag{13}$$

The parameters of elliptical motion contain the information on the fibre direction (azimuth angle $\theta$) and the degree of material reinforcement which is proportional to $\Delta\varphi$. To retrieve this information, one has to analyse the amplitude ($V_0$) and the phase ($\psi$) of the output signal of a similar receiving transducer as a function of its azimuth orientation ($\gamma$) [16,32]:

$$V_0 = \sqrt{V_1^2 + V_2^2 + 2V_1 V_2 \cos \Delta\varphi}; \ \psi = tg^{-1} \frac{V_2 \sin \Delta\varphi}{V_1 + V_2 \cos \Delta\varphi}, \tag{14}$$

where $V_1 = \cos(\beta - \theta) \cos(\gamma - \theta)$; $V_2 = \sin(\beta - \theta) \sin(\gamma - \theta)$.

To evaluate the fibre reinforcement, the four modes of operation are possible:

1.  Transmission mode (arbitrary orientations of the transmitter and receiver).
2.  Crossed transmitter–receiver orientation.
3.  Transmission for $\gamma = \beta$.
4.  Reflection birefringence mode.

In the first transmission mode, for a tetragonal composite (symmetrical in-plane reinforcement), $\Delta\varphi = 0$ and the material does not change polarization of the shear waves for any polarization angle $\beta$. The amplitude and phase of the output signal are calculated from (14) and shown in Figure 18 as functions of the polarization angle $\gamma$ of the receiver (for $\beta = 45°$ and $\Delta\varphi = 1°$). As one would expect, the output amplitude follows a $|\cos(\gamma - \beta)|$ relation and nullifies for crossed positions of the transmitter and receiver (figure-eight curve). The curves in Figure 18 (for $\Delta\varphi = 1°$) are, therefore, typical for "almost symmetrical" distribution of stiffness in the plane of a composite material. The asymmetry of the reinforcement is recognized by the polarization change to elliptical ($\Delta\varphi = 45°$), circular ($\Delta\varphi = 90°$) and the correspondent variations in the phases in Figure 18.

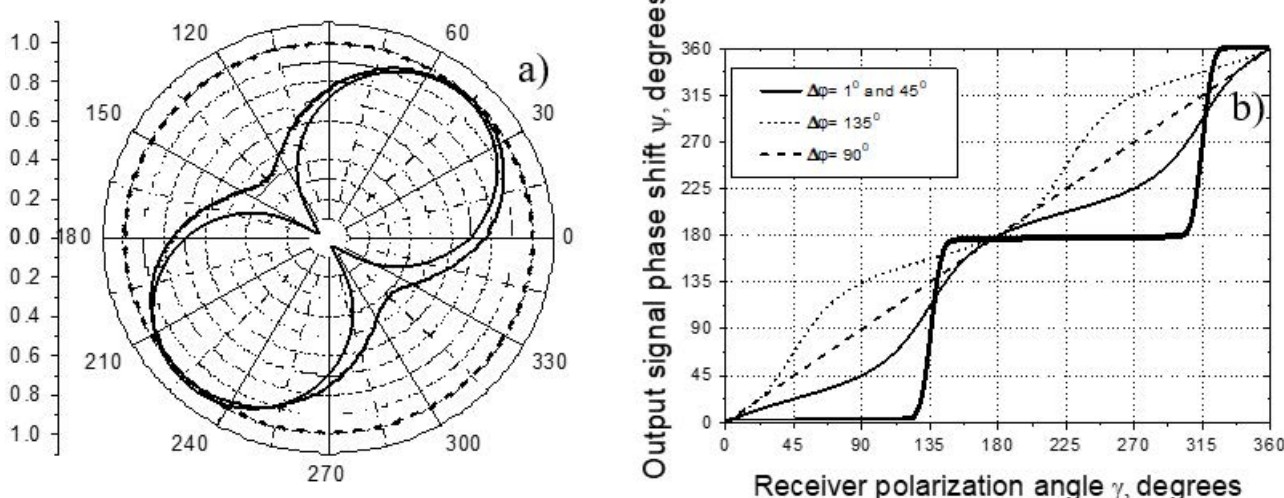

**Figure 18.** Output amplitude (**a**) and phase (**b**) as functions of receiver polarization angle for $\beta = 45°$: $\Delta\varphi = 1°$ and $45°$ (solid lines); $\Delta\varphi = 90°$ (dashed lines); $135°$ (dotted line in Figure 18b).

The crossed transmitter–receiver set-up corresponds to $\gamma = 90° + \beta$ in (14). Figure 19, a shows the results of calculations of the output amplitude $V_0(\gamma)$ in this mode for the fibres positioned at $\theta = 60°$. The fibre direction is recognized by a deep minimum of the output signal when a single wave polarized along the fibres is generated. As expected, the overall output amplitude decreases as the stiffness asymmetry ($\Delta\varphi$) reduces (inner curve in Figure 19a); it turns into zero in a material with fully symmetrical in-plane stiffness. Therefore, the crossed set-up is suited for discerning the fibre direction as well as the departure from isotropic lay-up in composite laminates.

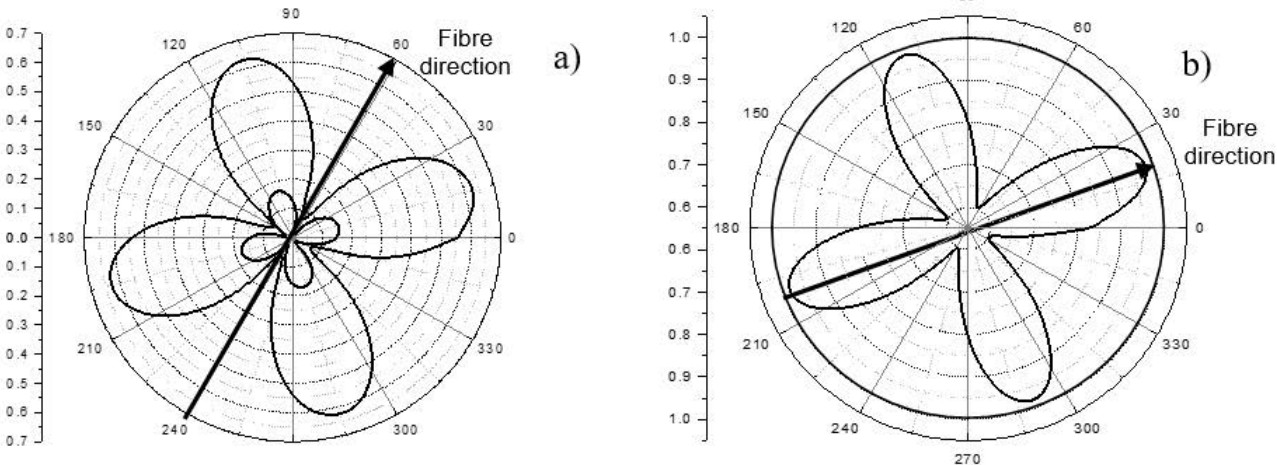

**Figure 19.** Calculations of $V_0(\gamma)$ in modes 2 (**a**) and 3–4 (**b**). (**a**) Fibre setup at $\theta = 60°$($\Delta\varphi = 80°$ for outward and $\Delta\varphi = 20°$ for inward curve); (**b**) parallel configuration with fibres at $\theta = 20°$ for $\Delta\varphi = 80°$ and $\Delta\varphi = 0°$ for outward circle.

For a parallel transmitter–receiver arrangement ($\gamma = \beta$) in transmission, the output amplitude peaks clearly at the fibre direction (Figure 19b). A single partial wave excited at this position is received in the best way by a parallel detector. The depth of the amplitude variation imprints the in-plane stiffness anisotropy: four-lobe response changes for a circle in a symmetrical lay-up. The parallel configuration is, thus, a sensitive tool for both the fibre orientation and in-plane stiffness anisotropy.

The condition $\gamma = \beta$ is automatically met in the reflection birefringence mode. Besides all the benefits of the parallel set-up, this mode uses a single transducer and provides a

single-sided access to the material. The latter is a very important factor for on-site material testing. For $\gamma = \beta$ from (14): max $|\psi| = |\Delta\varphi|$, i.e., the value of maximum phase shift directly quantifies the reinforcement asymmetry of in-plane stiffness. A "fast" polarization corresponds to a fibre direction and is readily revealed by a minimal signal delay. It is accompanied by a peak in the amplitude variation (Figure 19b) which also indicates the fibre orientation.

### 3.2. Partial Wave Approach: Sensitivity to Lay-Up Inconsistency

Unlike lattice anisotropy of crystalline materials, in laminate composites, the stiffness anisotropy is formed by a lay-up of plies. Each of them displays a twofold symmetry but a resulting anisotropy is a function of the number of plies and their orientation. Therefore, such an artificial anisotropic structure can be represented as a superposition of orthotropic layers. As it was shown above, a shear wave entering each of arbitrarily oriented birefringent layers is decomposed into two shear waves which also acquire an additional phase shift. After traversing through a laminate composite, a superposition of all partial waves forms the resultant wave field measured by a receiving transducer.

The feasibility of such an algorithm is illustrated below for the parallel transmission and reflection birefringence modes in a multiply $(0° + 45° - 45° - 90°)$ CFRP laminate (Figure 20). In the first $(0°)$ layer, the radiated shear wave is split into two waves ("fast" and "slow") polarized along the x- and y-axes so that the input of $(+45°)$ layer is: $U_1 = U_0 \cos\alpha \sin(\omega t + \delta_1)$ and $U_2 = U_0 \sin\alpha \sin(\omega t)$, where $\delta_1$ is the phase shift due to difference in the wave velocities ($v_{fast}$ and $v_{slow}$). In the next layer, each of these waves is decayed into a similar pair of partial waves which acquires a relative phase shift $\Delta_1$. In the $(-45°)$ area, no wave decomposition takes place but the pairs of the "fast" and "slow" waves change place and acquire additional phase shift $\Delta_2$. After a final decomposition in the $90°$ (x″, y″) layer (phase shift $\delta_2$), four pairs of waves are summed up by the receiver whose output amplitude and phase are: $U_{out}^2 = \sum\limits_{i=1}^{8}(U_i \sin\psi_i)^2 + \sum\limits_{i=1}^{8}(U_i \cos\psi_i)^2$;

$\psi = tg^{-1}\left(\sum\limits_{i=1}^{8}(U_i \sin\psi_i)/\sum\limits_{i=1}^{8}(U_i \cos\psi_i)\right)$, where $\psi_i$ the various combinations of $\delta_{1,2}$ and $\Delta_{1,2}$.

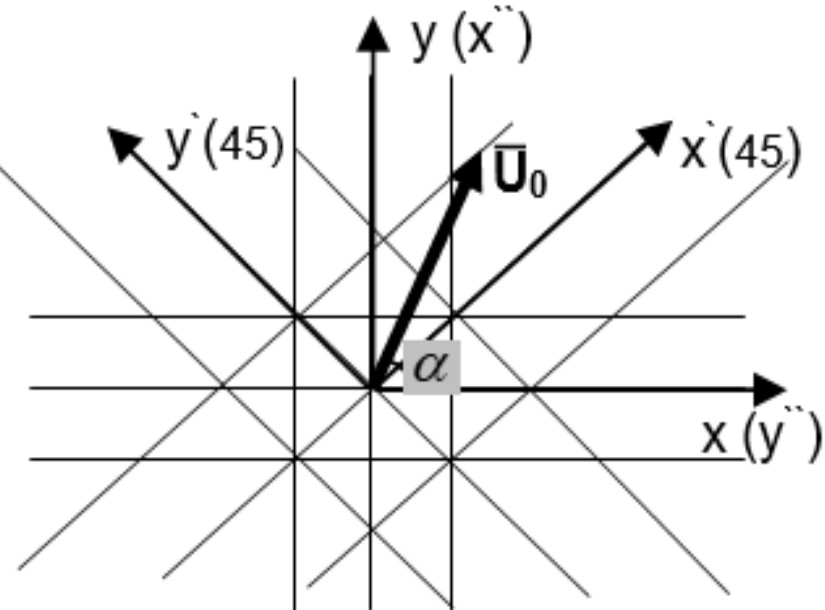

**Figure 20.** Birefringence in $(0° + 45° - 45° - 90°)$ composite laminate.

It is instructive, to demonstrate the two cases of symmetrical lay-ups: $\Delta_1 = \Delta_2$ and $\delta_1 = \delta_2$. For a symmetrical $(+45° - 45°)$ lay-up $(\Delta_1 = \Delta_2 = \Delta)$, the above relations are simplified:

$$U_{out} = U_0 \sqrt{\sin^4 \alpha + \cos^4 \alpha + 2 \sin^2 \alpha \cos^2 \alpha \cos(\delta_1 - \delta_2)}$$

$$\psi = tg^{-1} \left( \frac{\sin(\delta_2 + \Delta) \sin^2 \alpha + \sin(\delta_1 + \Delta) \cos^2 \alpha}{\cos(\delta_2 + \Delta) \sin^2 \alpha + \cos(\delta_1 + \Delta) \cos^2 \alpha} \right)$$

The results of calculations (Figure 21) show that azimuth angle variations of both the amplitude and the phase depend exclusively on the in-plane stiffness asymmetry $(\delta_1 - \delta_2)$ between the 0° and 90° layers. The contribution of the symmetrical $(+45° - 45°)$ lay-up is zero due to balancing of birefringence. When $(\delta_1 - \delta_2) = 0$, the birefringence in (0–90°) lay-up is also cancelled and the composite laminate is fully isotropic.

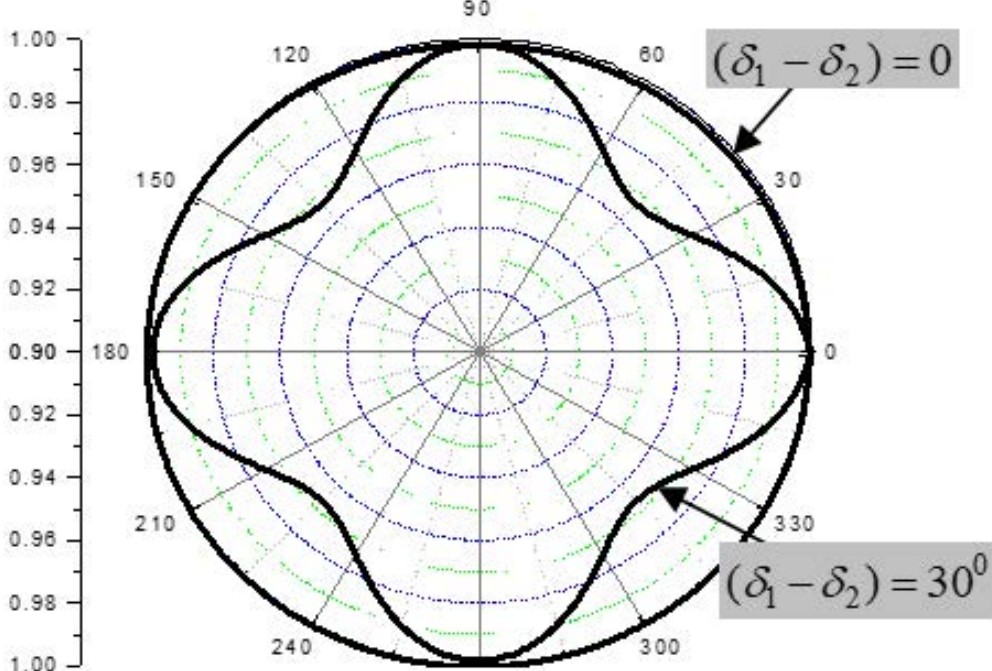

**Figure 21.** Amplitude response of birefringence to ≈60 μm offset in 0–90° CFRP lay-up.

Thus, in the cross-plied composite laminate, the birefringence probes the difference in the in-plane stiffness between successive plies. That provides an extraordinary high sensitivity of the technique to any deviation from the symmetry caused either by extra plies or change in their alignment.

As an example, Figure 21 also shows the output amplitude and phase variations for the (0–90°) lay-up which is "unbalanced" by $\delta_1 - \delta_2 = 30°$. For the birefringence in UD-CFRP composite [16], this phase difference corresponds to an offset in thickness of ≈60 μm between the 0° and 90° layers. Such an imbalance of less than a half-ply thickness can be revealed in the deviation of the amplitude curve (Figure 21). The fibre orientation in the extra ply thickness (0° or 90°) is readily indicated by a "fast" polarization direction of the transducer.

In a similar way, a minor asymmetry in the $(+45° - 45°)$ lay-up can be detected readily on the background of a symmetrical 0°–90° structure (Figure 22). The ±45° rotational turn of the amplitude curve indicates the source of the offset. The phase measurements enable to quantify the inaccuracy; the "fast" polarization direction of the transducer specifies particular (+45° or −45°) orientation of an extra ply.

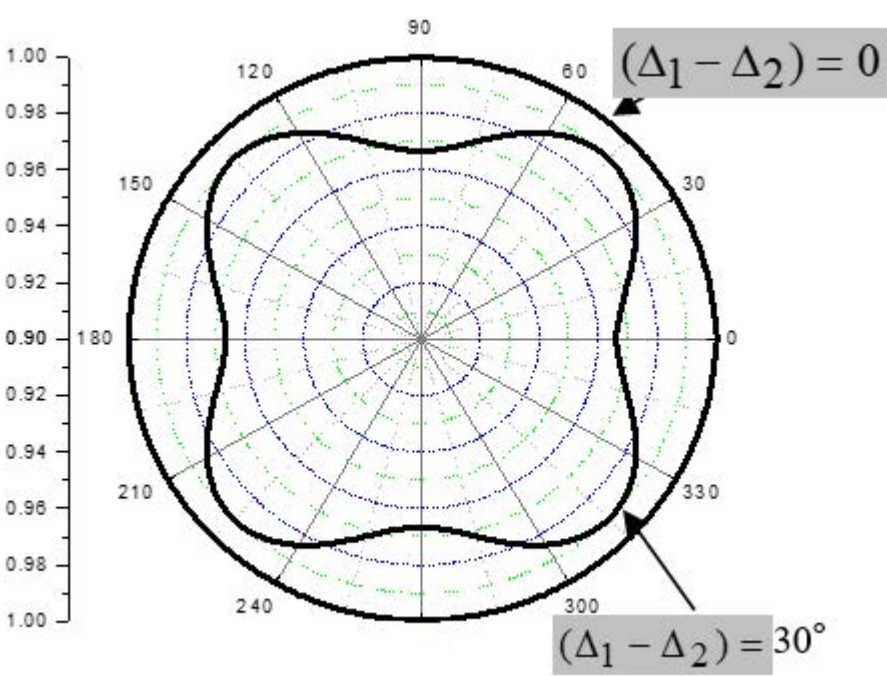

**Figure 22.** Amplitude response of birefringence to ≈60 μm offset in +45 − 45° CFRP lay-up.

## 4. Birefringence Applications

### 4.1. Mapping of Fibre Fields in Composites

As it was shown in Section 3.1, the fibre direction in a unidirectional (UD) reinforced composite is readily determined in the birefringence reflection mode by detecting a "fast" wave polarization. Such an experiment includes a single sided shear wave generation/detection followed by the measurements of the amplitude/phase (delay) of the output signal as a function of polarization angle (polarization amplitude/phase (velocity) curve). In our experiments, the Krautkrämer ultrasonic flaw detector (USIP 12) was used for excitation/reception of (2.5–4) MHz shear wave pulses generated by piezo-transducers with known polarization direction. The transducer was attached to the hand of the six-axis robot IRB 120 (supplier ABB, Switzerland) and pressed against the specimen through a layer of viscous ultrasonic coupler; the generated/detected wave polarization was varied by rotation of the transducer (Figure 23). Such a setup is sufficient for mapping the fibre orientation by recording the polarization angle corresponding to the "fast" wave in the specimen. An example of the polarization velocity curve measured in a stacked UD GFRP is shown in Figure 24 where the "fast" wave polarization is along 0° and indicates the orientation of continuous fibres in the plate.

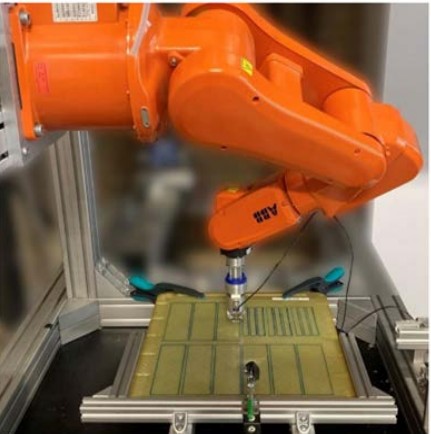

**Figure 23.** Experimental setup for birefringence measurements and applications.

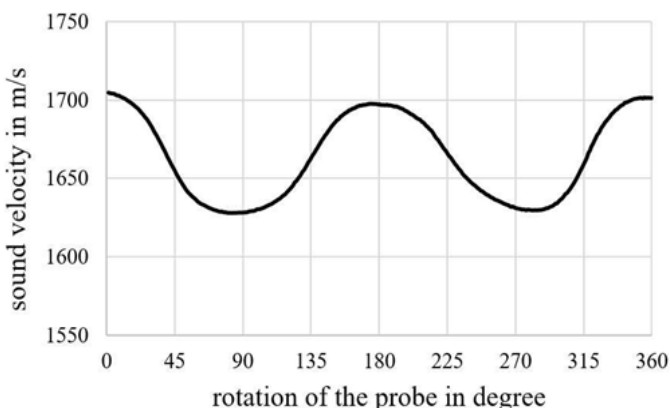

**Figure 24.** Polarization velocity curve measured in a UD GFRP plate.

The examples of mapping of short glass fibre orientations in polymer industrial components produced by injection moulding are shown in Figure 25. The "fast" polarization directions are indicated by arrows; the length of the arrows is proportional to a local degree of reinforcement ($\Delta\varphi$). The specimen in Figure 25a is a large ($300 \times 200 \times 4$ mm³) polyurethane plate with the holes formed by circular barriers in the mould. The arrow pattern measured is in a good agreement with expected streamlines in the mould (polymer flow from right to left). Some additional reinforcement observed around the holes is evidently due to accumulation of fibres in the vicinity of the barriers.

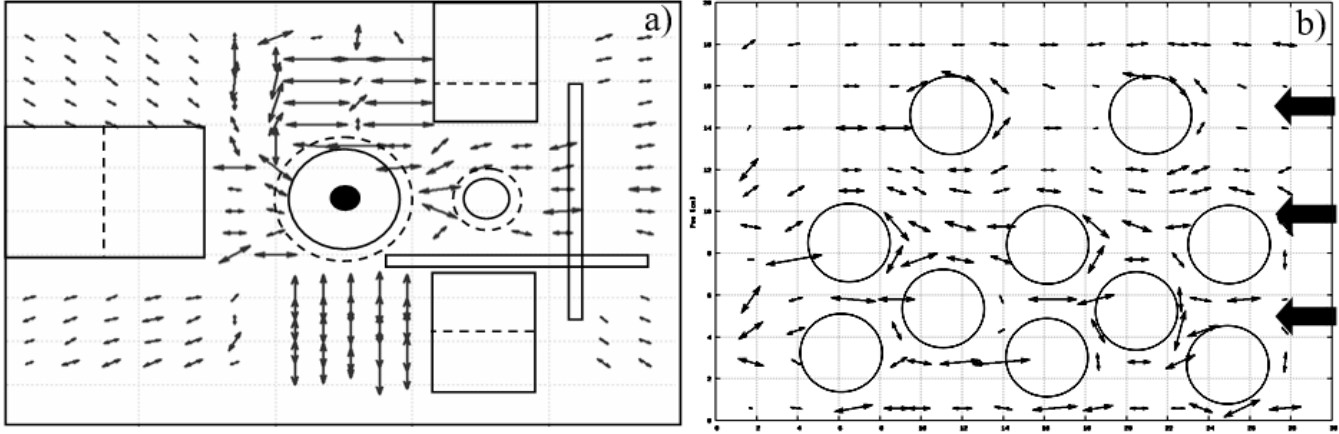

**Figure 25.** Fibre reinforcement mapping in injection molding specimens of GFRP: polypropylene specimen $160 \times 250 \times 2$ mm³ (**a**); polyurethane $300 \times 200 \times 4$ mm³ (**b**) specimen.

The polypropylene specimen in Figure 25b has a complex shape with a series of horizontal and vertical stripes separated by narrow slots. An overall radial arrow pattern is due to a central position of a polymer flow inlet. A drastic enhancement of the fibre alignment is observed in the areas of strips and caused by regularization of the flow around the barriers.

Similar measurements also enable to trace particular fibre (or fibre bundle) directions that could be applied for recognition of a characteristic composite pitfall: an in-plane fibre undulation. For 2D-parallel continuous-fibre structures, the local fibre orientation (angle $\alpha_i$) represents a derivative $dY/dX = tg\alpha_i$ of the fibre trajectory ($Y(X)$) in the point of measurement ($X_i$). As a result, a series of $\alpha_i$ measurements in the point along the X-axis enables to reconstruct the fibre trajectory by using the following relation:

$$Y(X) = Y_0 + \sum_{i=0}^{n} \Delta x tg\alpha_i, \tag{15}$$

where $Y_0$ is the initial fibre coordinate and $\Delta x$ is the distance between the measurement points.

Relation (15) was applied for probing a fibre trajectory in a unidirectional CFRP specimen with specially produced in-plane undulation areas. The results in Figure 26 show that even a small (5°) deviation in the fibre alignment is detected reliably by using the birefringence technique.

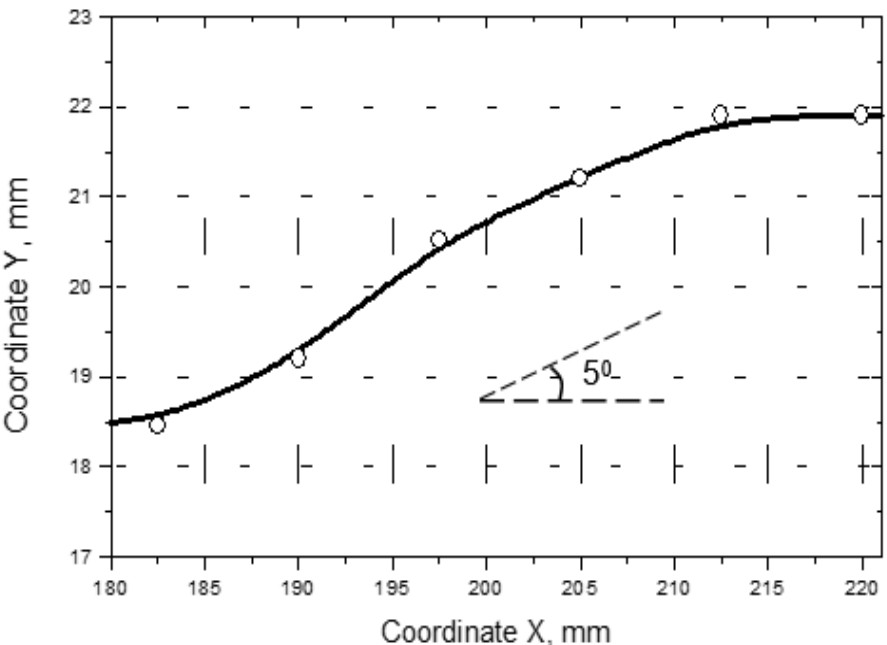

**Figure 26.** Reconstruction of fibre trajectory in a 5° in-plane undulation area of a CFRP specimen.

In multiply composites, a similar experiment identifies a major reinforcement direction which is caused by various fibre orientations in lay-up stacks. To identify the fibre layups involved the measured polarization velocity curves have to be examined against the simulation results for reference composite structures. The latter can be obtained by using the analytical partial wave approach (Section 3.2) or the MatLab calculations of successive reflections/transmissions at the interfaces between the plies [33]. The minimum difference between the measured and the simulated results indicate a close equivalence of the multiply composite studied to the reference arrangement.

Figure 27 shows the measured and simulated results for velocity polarization curves (polar plots) obtained in GFRP composites with various fiber lay-up stacks [34]. The simulation was performed by using the MatLab simulation approach [33]. The measurements were carried out by using the 1 MHz shear wave transducer. The signal for the transducer was a sinus burst of one cycle. Polarization angle scans were performed in 1° steps that were also used to calculate the phase velocity. The impact of the extra plies on top of 0°-plies is intuitively clear: the main reinforcement direction (max velocity) departs progressively from 0° direction (Figure 27).

To estimate the agreement between the experiment and the simulation, the RMS error is calculated for each set of measurement which is found to be <1%. Thus, the simulation procedure recognizes the lay-up stacks quite precisely. If, therefore, a gap between the measured and simulated polarization velocity curves is found to be substantially greater, it might be an indication of faulty lay-ups or some other manufacturing-related defects caused by the fiber orientation.

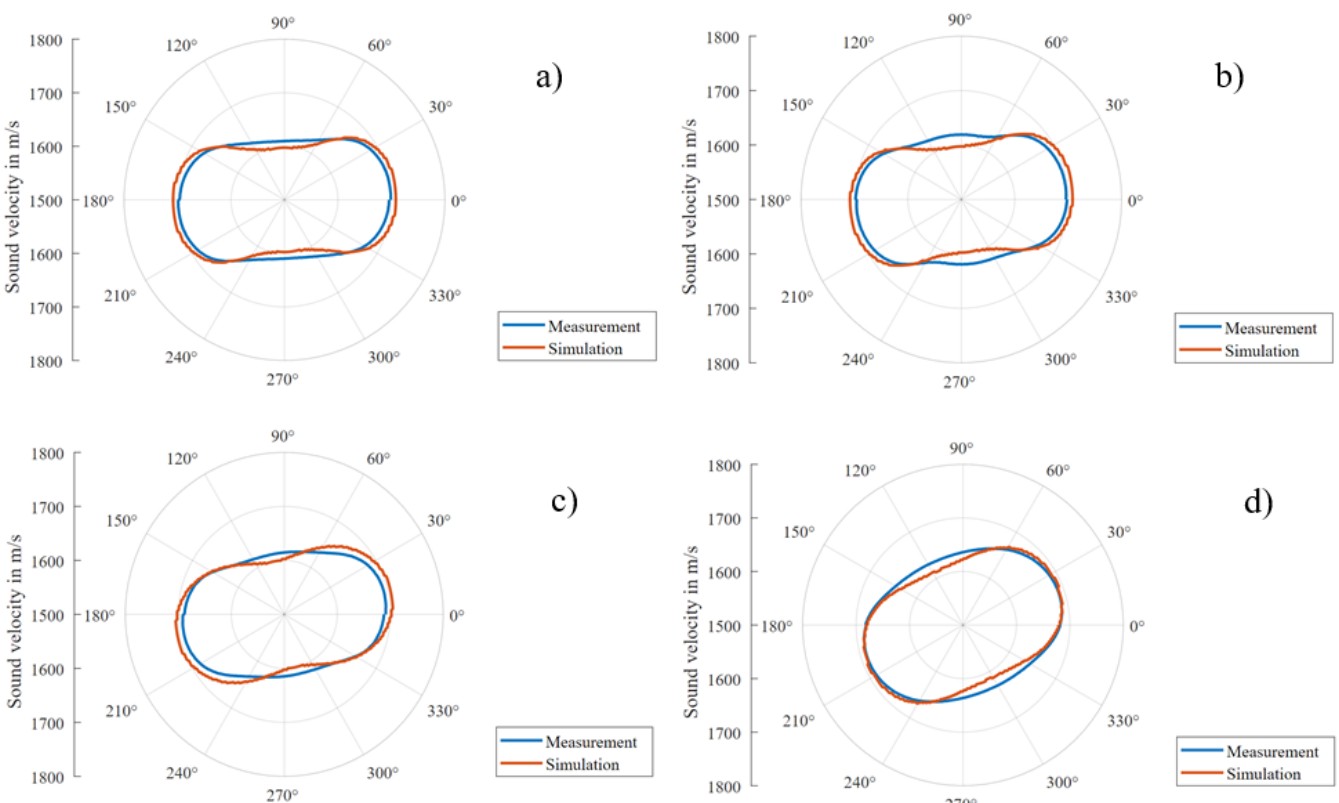

**Figure 27.** Comparison between measured and simulated polarization velocity curves for various fibre stacks in GFRP: (**a**) [0°/0°/5°/5°], (**b**) [0°/0°/10°/10°], (**c**) [0°/0°/20°/20°] plate, (**d**) [0°/0°/45°/45°] plates.

### 4.2. Monitoring Damage Development in Composites

Besides being important for choosing the material orientation to respond to in-service loads, the "stiff" and "soft" directions are relevant for the development of damage in composites, e.g., induced by impacts [35,36]. The crack opening is shown to be more easily produced in a weakly bonded direction so that impact-induced cracks propagate, predominantly, along a strongly bonded (reinforcement) orientation (Figure 28a). Since the in-plane shear stiffness conforms to the binding force anisotropy, the birefringence can be used to predict orientation of cracking in composite materials. In the bi-axial composite (Figure 28b), the cracking area is strongly elongated along the 90° axis, so that predominant crack orientation also corresponds to the direction of the higher shear stiffness.

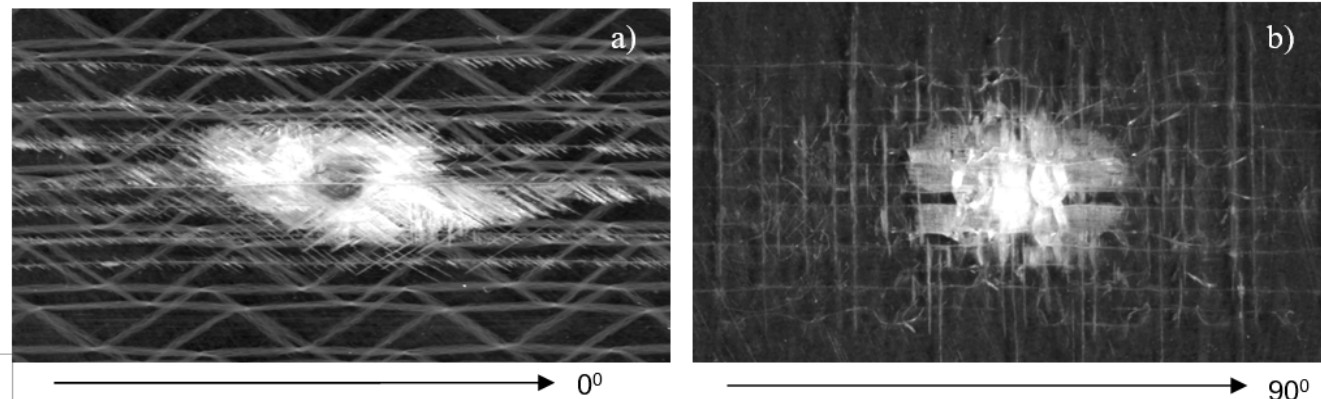

**Figure 28.** Impact-induced cracking predominantly along stiff the 0° and −45° directions in 4-axial (**a**) and bi-axial (**b**) fabric composites.

Damage and cracking produced in composites obviously modifies its in-plane stiffness anisotropy and affects the birefringence pattern. To verify the feasibility and sensitivity of the ultrasonic phase variations, the measurements were implemented for a series of bi-axial fabric composite specimens with the damage induced by tensile loading. The results of output phase variations obtained by rotating the 2.25 MHz-shear wave receiver in transmission mode are shown in Figure 29b for different tensile loads applied. The step-wise phase behaviour measured for the 4 kN case indicates that material is weakly anisotropic. Comparison of the experimental data with calculations from (14) confirms that the 90° axis is slightly stiffer in shear with $\Delta\varphi = 5°$. For higher loads, the stiffness anisotropy pattern changes: the asymmetry in shear stiffness between the 90° and 0° directions increases, making the material more anisotropic. The phase shift bounds from $\Delta\varphi = 10°$ for 8.1 kN $\Delta\varphi = 30°$ for 16 kN load. Such an increase correlates well with macro-cracking induced in the specimen in this range of loads and aligned along the stiffer 90° direction.

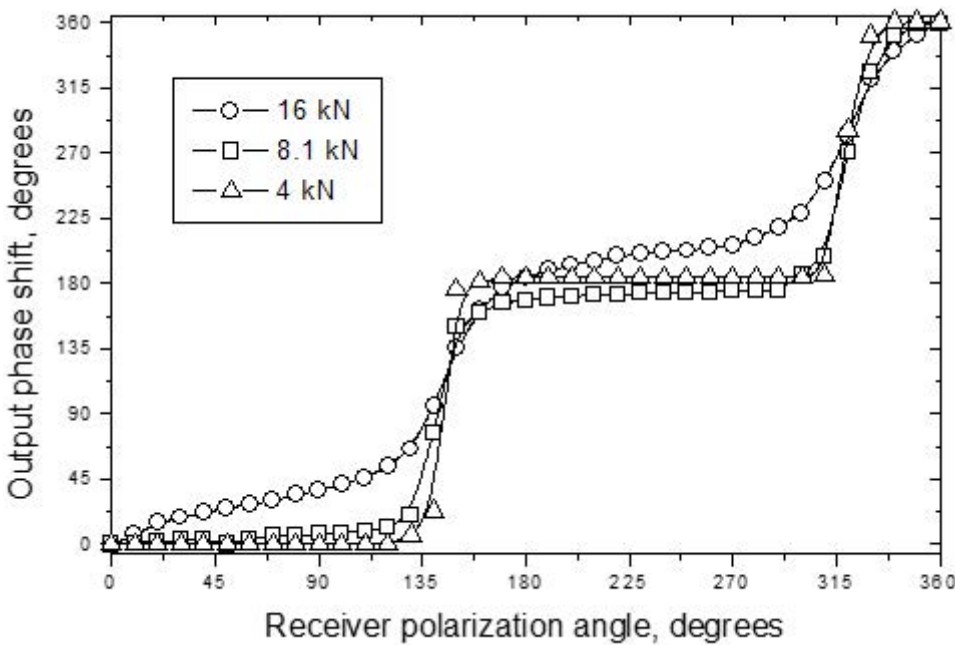

**Figure 29.** For different tensile loads applied to bi-axial (0–90°) glass fabric composite.

## 5. Conclusions

In the ultrasonic frequency range, the stiffness anisotropy assessed by using bulk wave velocities is independent of frequency and fully determined by elastic moduli of composite material. In plate-like composites, the in-plane velocity anisotropy measured for plate waves is not constant but is a function of frequency even for materials deemed to be homogeneous and non-dissipative. The reason for that is concerned with velocity dispersion, which affects the wave propagation in different ways for various azimuth directions and thus modifies the elastic anisotropy pattern. Such an effect of dispersion of elastic anisotropy is of importance in dynamic testing and elastic wave applications in composite materials. It also provides the dispersion of beam steering phenomena in composites that causes energy focusing in the neighbourhood of the fibre direction ("phonon focusing") in strongly anisotropic composites.

In bi-directional fibre-reinforced composites, a linearly polarized shear wave is decomposed into two shear waves polarized along the reinforcement directions and propagating with different velocities (acoustic birefringence). The parameters of elliptical motion induced by the birefringent ultrasonic shear waves in fibre-reinforced composites deliver information on the in-plane stiffness anisotropy of the material. A reliable compliance between the theoretical description developed and the experimental data enable to estimate the strength of birefringence and to derive the stiffness asymmetry for shear strain. The

birefringence approach is suitable for rapid non-destructive mapping of the short- and long-fibre fields in both injection molding and multiply composites. The testing method involves standard ultrasonic instrumentation (burst mode) and is based on the measurements of the shear wave delay as a function of the polarization angle (polarization velocity curve). The measurements are readily automated by using robots for the variation of the wave polarization by the shear wave transducer rotation. The technique also quantifies the in-plane stiffness anisotropy and identifies various fibre orientations in lay-up stacks. Cracked damage in composite materials modifies the stiffness anisotropy pattern and is also revealed by the birefringence measurements.

**Author Contributions:** Conceptualization, I.S. and M.K.; methodology and validation, I.S., Y.B., L.L.; All authors have read and agreed to the published version of the manuscript.

**Funding:** This research was funded by German Research Foundation (DFG); Grants numbers: 5332680; 26724186; 428323347.

**Data Availability Statement:** The data presented in this study are available in the authors' articles in the list of references.

**Acknowledgments:** The authors are grateful to the German Research Foundation (DFG) for funding this work as part of the following projects: 5332680; 26724186; 428323347. The authors would also like to thank the ILK of the TU Dresden for the production and provision of the test samples.

**Conflicts of Interest:** The authors declare no conflict of interest.

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
