# Peer review of "Ultrasonic Anisotropy in Composites: Effects and Applications"

_jcs, doi:10.3390/jcs6030093_

Round 1

Reviewer 1 Report

please review comments provided in the attachment

Author Response

Replies to Reviewer 1 comments.

Comment:

Please, include some of the latest developments in guided wave ultrasonics.

Reply: Accepted and revised. See a new list of references.

Comment:

Please, cross reference all references.

Reply: The comment on making cross references is more like a technical question/task to Editorial Team whether they consider it necessary for the publication in this journal.

Comment (p.3):

Please explain or show this explicitly

Reply: Accepted and revised (p.3)

Comment:

CFR

Reply: Accepted and revised.

Comment:

Please, name the individual figures as (a) and (b).

Reply: Accepted and revised for all Figures throughout the manuscript.

Comment:

Please, show the axis labels in Fig. 4.

Reply: Accepted and revised.

Comment:

CFRP abbreviation is incorrect.

Reply: Accepted and revised.

 Comment:

Explain velocity anisotropy in Fig. 6.

Reply: Accepted and explained.

Comment:

Please, give legend of what solid line vs circles is in Fig. 11.

Reply: Accepted and revised.

Comment:

Please, give legend of what solid line vs circles is in Fig. 12.

Reply: Accepted and revised.

Comment:

Please, explain the curves order in Fig.13.

Reply: It is explained clearly in the Figure caption. The arrow and the caption clearly refer to the curve order.

Comment:

Please, add space between the two words (p.12).

Reply: Accepted and revised.

Comment:

(p. 13) Why didn`t use the same fd values as in Fig. 15?

Reply: Calculations in Fig. 15 used the ultimate values fd just to illustrate possible behaviour in those cases. The experiment is limited by the transducer frequency response which prevented from using exactly the same values.

Comment:

What is the meaning „figure-eight“ (p. 15)?

Reply: Figure-eight means a figure shape like number 8.

Comment:

To add legends in Fig.18.

Reply: The legend is added to Fig. 18, b in the free space available. The number of curves in Fig. a is reduced to enhance the clarity.

Comment:

Fig. 19 is extremely confusing.

Reply: The number of curves is reduced in the Fig. to enhance the clarity.

Comment:

(p.18) 60 m misprint.

Reply: Revised.

Comment:

(p.22) Include recent references on impact damage detection.

Reply: Accepted and revised. See ref. 35, 36 in a new list of references.

Comment:

(p.23, Fig. 29) Not the right way to detect this type of damage.

Reply: This just an illustration of possible application of birefringence for damage detection. We never state that this is the best way….

Reviewer 2 Report

 The authors have delivered  extensive and thus valuable discussion both theoretic as well as practical aspects of the topic. The issues are presented clearly. The single inconvenience of the paper is mixing of the theoretical introduction (large of necessity)  with experimental results. To  improve the presentation of the topic the referee suggests inserting of the short  summary of recommended testing methods within the section of Conclusions.

Author Response

Reply to Reviewer 2 comment.

Comment: To improve the presentation of the topic the referee suggests inserting of the short summary of recommended testing methods within the section of Conclusions.

Reply: Accepted and revised. Corresponding paragraph is included in the Conclusion section.

Round 2

Reviewer 1 Report

This version of the manuscript is greatly improved based on the recommendation of the reviewers